# Artificial Neural Network Structure Optimisation in the Pareto Approach on the Example of Stress Prediction in the Disk-Drum Structure of an Axial Compressor

**DOI:** 10.3390/ma15134451

**Published:** 2022-06-24

**Authors:** Adam Kozakiewicz, Rafał Kieszek

**Affiliations:** Faculty of Mechatronics, Armament and Aerospace, Institute of Aviation Technology, Military University of Technology, 00-908 Warsaw, Poland; adam.kozakiewicz@wat.edu.pl

**Keywords:** artificial neural networks, optimization, genetic algorithm, turbine engines, axial compressors, predictive model

## Abstract

The article presents the process of selecting and optimising artificial neural networks based on the example of determining the stress distribution in a disk-drum structure compressor stage of an aircraft turbine engine. The presented algorithm allows the determination of von Mises stress values which can be part of the penalty function for further mass optimization of the structure. A method of a parametric model description of a compressor stage is presented in order to prepare a reduced stress distribution for training artificial neural networks. A comparative analysis of selected neural network training algorithms combined with the optimisation of their structure is presented. A genetic algorithm was used to determine the optimal number of hidden layers and neurons in a layer. The objective function was to minimise the absolute value of the relative error and standard deviation of stresses determined by FEM and artificial neural networks. The results are presented in the form of the Pareto front due to the stochastic optimisation process.

## 1. Introduction

Artificial neural networks have been known to artificial intelligence researchers for over 70 years. W. McCulloch and W. Pitts developed the first formal description of the artificial neuron model in 1943, presented in paper [1]. The mathematical model of the McCulloch–Pitt neuron [1] is described in terms of the Equation (1):(1)$$y_{i}=f\left(u_{i}\right)=f\left(\overset{N}{\underset{j=1}{\sum }}w_{ij}x_{j}\left(t\right)+w_{i0}\right)$$
where: 

$u_{i}$—argument of the summation function;

$w_{ij}$—value of the selected weighting factor;

$x_{j}\left(t\right)$—input signal to the neuron;

$w_{i0}$—value of bias.

As shown in the work of [2], the sigmoidal threshold function gives the best results described by the Equation (2).
(2)$$f\left(u_{i}\right)=\frac{1}{1+\exp\left(-\beta u_{i}\right)}$$
where: 

β—steepness coefficient of the function.

In Equation (1), the weights $w_{ij}$ can take any value. The case when $w_{ij}>0$ means excitatory connection, when $w_{ij}<0$ means suppressive connection, and when $w_{ij}=0$ means no synaptic connection. The adoption of a discrete neuron model is justified by the phenomenon of refraction. This means that a biological neuron can only operate at a certain frequency.

In his book [3], D. Hebb discovered that the McCulloch–Pitt neuron can store information in its structure using weight values. In addition, he proposed a way to teach the network by manipulating the weight values. He derived his idea of teaching neurons from observations of nature. He observed that the weight of connections between nerve cells is amplified when they are activated at the same time. That is, the more often a connection is used, the stronger it becomes. Mathematically, Hebb’s model is represented as follows:(3)$$w_{ij}\left(k+1\right)=w_{ij}\left(k\right)+\eta y_{i}\left(k\right)y_{j}\left(k\right)$$
where: 

k—cycle number;

η—learning rate;

$y_{i},y_{j}$—output value from the *i*-th and *j*-th neuron.

In recent years, the possibility of combining Artificial Neural Networks (ANNs) with hard algorithms for numerical calculations is being studied. The aim is to reduce the time of numerical calculations, which is directly related to the design costs. An analysis of selected works presenting the problem of applying neural networks for optimisation using hard algorithms is presented below [4,5,6,7].

ANNs are increasingly being used in aviation [8,9,10]. They are used in reliability analysis [11,12], fault detection [13,14,15,16], identification [17], control [18,19], and design and optimisation [20,21,22,23,24,25].

Pierret and Van den Braembussche, in their work [26], presented the results of compressor palisades and turbines optimisation, using an algorithm containing an ANN. The authors were able to reduce the pressure loss coefficient by about 4% in relation to the output system. The algorithm proposed by the authors can be successfully used to optimise turbine or compressor blades, for both profiles alone and in 3D systems. The algorithm excludes the analysis of unreal blades, and the use of the ANN significantly increases the speed of calculations, which is a major advantage. Moreover, the applied algorithm made it possible to fully automate the optimisation process, ensuring that the algorithm does not indicate a local minimum, very distant from the global one.

In an article by Keshtegar [27], an ANN was used to predict stresses, displacements, and strains in turbine engines. The authors analysed selected parameters during a given mission in a BLISK turbine disc.

In the work [28], similarly to the works [29,30], the reliability of the turbine rotor in terms of creep was assessed for the clearances. The authors decomposed the rotor model into component models (i.e., disc, blades, and body), which were analysed using FEM. The data were used to train a neural network, which was able to achieve an RMSE accuracy of $6.32\times 10^{-4}$ for the worst matching disc component. As a result, an overall solution accuracy of ~99.93% was achieved for 104 simulations. The authors believe that the demonstrated algorithm presents a promising approach in the field of dynamic reliability analysis of complex structures.

In the publication [31], an innovative method of temperature field predicting in a three-dimensional model of a turbine rotor rim was presented. For this purpose, an algorithm combining the FVM (Finite Volume Method) and an ANN was used. Using the ANSYS program, the temperature distribution was determined, which serves as input data for ANN training. This was able to predict the temperature distribution for selected nodes of the turbine model with the change of rotor rotational speed, temperature, and inlet pressure. The maximum absolute temperature error was 44.7 K, and mean absolute error (MAPE) was 0.5%. A similar problem was solved in paper [31] where the main goal was minimisation time of temperature prediction. Elapsed time using the CFD method is one h, 42 min, and 44.7 s, and elapsed time using the ANN method was 0.0027645 s.

Stress analysis using an ANN in wind turbine blades is the subject of papers [32] and [33]. The first study uses Genetic Algorithm–Back Propagation (GA-BP) and the second Particle Swarm Optimization–Back Propagation (PSO-BP). In both cases, a relative error of about 6% (on the leading edge, on the overpressure side) was achieved compared to full-scale static tests. For the trailing edge, the relative error was 6.5% and 18% for the PSO-BP and GA-BP algorithms, respectively. There are also other examples of ANN use available in the literature, e.g., in wind turbine optimisation [34,35]. There are also works questioning the validity of approximating the objective function with an ANN [36], which is a counterexample to the previously cited works.

In the analysed area of the literature, the authors did not come across any publications related to the use of an ANN in disk systems or disk-drum turbine engines optimisation. Previous attempts of the authors to use an ANN in compressor disc calculations are presented in publications [9,10]. This paper is an extension of these works, aimed at optimising and extending the research of the previously used algorithm.

The article presents the results of applying an ANN to reduce the time of determining the penalty function value in the case of the structural system strength analysis of the compressor stage. The constructed model was based on a disk-drum system of a turbojet engine compressor. The stresses necessary to train the ANN were obtained using the APDL language. Additionally, the network structure was optimised due to the relative error and standard deviation of stress prediction, and the solutions were presented in the form of the Pareto front. In the classic Pareto approach, two objective functions are optimised, which are, for example, related to weight values [37], or a solution is sought in the solution space using, for example, the cone separation method [38]. In the article, the optimal solutions front was determined by the simultaneous minimisation of two objective functions using the stochastic method, and therefore, the Pareto front was outlined on the basis of many stochastically optimal solutions.

## 2. Use of the APDL Language to Determine the Disk Stresses

The greatest advantage of APDL (Ansys Parametric Design Language) is its adaptation to parametric modelling, which provides a set of solutions. Using the APDL language, it is also possible to discretise the model, set parametric boundary-initial conditions and perform numerical analyses. It is also possible to generate paths along which, regardless of the model variant, the selected results of numerical analysis are interpreted. Due to these advantages, it was decided to use the APDL language. The aim of the work was to create a parametric model of the disk-drum compressor structure, an aircraft turbine engine, whose single stage is shown in Figure 1. The parametric model of the stage is shown in Figure 2.

The model shown in Figure 2 was constructed to ensure the tangency of the curves (G1 connection class) at the points of the system element connections, with the exception of the hub-drum connection, where the G0 connection class was used, i.e., no derivatives continuity. The model used is axisymmetric to the vertical axis (Figure 3), as the APDL language structure requires this.

The discretisation of the system was carried out using the “AMESH” function with a set mesh size of 0.005.

Based on Equation (4) [39] in the polar coordinate system, summing the forces in the radial direction, including body force R per unit volume in radial direction and dividing by dr dΘ, we obtain the equation for equilibrium:(4)$$\frac{{\left(\sigma_{r}r\right)}_{1}-{\left(\sigma_{r}r\right)}_{3}}{d\Theta }-\frac{1}{2}\left[{\left(\sigma_{\Theta }\right)}_{2}+{\left(\sigma_{\Theta }\right)}_{4}\right]+\frac{{\left(\tau_{r\Theta }\right)}_{2}-{\left(\tau_{r\Theta }\right)}_{4}}{d\Theta }+Rr=0\;$$
where: 

$\sigma_{r}$—radial stress component;

$\sigma_{\Theta }$—circumferential stress component;

$\tau_{r\Theta }$—shearing stress components;

R—body force per unit volume in radial direction;

r—radius.

Index 1, 3 is normal to the plate, and 2, 4 is radial direction.

For axisymmetric stress distribution, stress components do not depend on Θ. After a few simple transformations, the end of the disc (Figure 3, B) is loaded by pulling according to the Equation (5) [40,41,42]. This equation binds centrifugal force from blade and lock mass and centrifugal (radial) stress on the outside surface of the disc.
(5)$$\sigma_{w}=\frac{\left(F_{t}+F_{z}\right)\;n}{2\pi \;r_{max}\;h_{min}}\;$$
where: 

$\sigma_{w}$—disc centrifugal stress;

$F_{t}$—centrifugal force from the lock;

$F_{z}$—centrifugal force from the blade;

n—number of blades;

$r_{max}$—outer radius of the disc;

$h_{min}$—thickness of the disc.

The pull was performed using the SFL (Specifies Surface Loads) function. The rotation speed of the disc system is set using the OMEGA function. The symmetry condition is set at the point where the drum is connected to the next stage drum (in Figure 2, zones A and B). This approach makes it possible to decompose the analysis of the entire compressor rotor (Figure 4) into individual stages. It was assumed that the connection of the stages is sufficiently rigid, and the symmetry condition in this area does not introduce significant errors.

Solving the numerical problem, three paths were generated in the post-processor along which the data necessary for the calculations were interpreted. Each path is defined by two points. The first path is based on the centre of the hub base and the centre of the disc tip (Figure 2—blue line); the second runs through the centre of the left drum section (Figure 2—red line), and the third through the centre of the right drum section (Figure 2—green line). This made it possible to obtain the stress distribution calculated in accordance with the hypothesis of the maximum energy of shear deformation (Huber–Mises–Hencky theory) in the sections necessary for the profiling of the structure. An example of the stress pattern for the first path is shown in Figure 5.

The disc section path was divided into twenty nodes and the drum section path into five. The created numerical model was validated using the stochastically selected geometric parameters of the model and loads. The results of the numerical analysis were compared with the analytical calculations bringing the model geometry closer to the straight disc. This convergence is achieved by reducing the thickness of the hub to 5.1 mm at the disc thickness of 5 mm, reducing the drum thickness to 0.5 mm and the $R_{i}$ radii of curvature to 5 mm. It can be seen that the obtained model corresponds to the straight disc arrangement. The stress distribution for the analytical model ($\sigma_{AN}$) and APDL ($\sigma_{APDL}$) and relative error (δ) for the obtained models are presented in Figure 6.

For r=0.4 m, the greatest error occurred due to the presence of an additional structural element in this area (due to the disk–drum connection). For the remainder, the error does not exceed 0.6%.

In a further part of the paper, the APDL algorithm is used to generate the input training parameters described in Figure 2: *r_max*—radius of disk, *r_min*—radius of central hole, *h*_0_—hub width, *h_min_*—disc width, *lbl*—length of left drum, *rbl*—radius of left drum, *lbr*—length of left drum, *rbr*—radius of left drum, and *rb*—radius of location of the centre of the circle round the disc-hub joint. Additionally, there are a few parameters connected with material properties and blade geometry such as: Young module, density, Poisson’s ratio, mass of single blade, number of blades, radius of the centre of mass of the lock, radius of the centre of mass of the blade, volume of lock, and rotational speed. The Von Mises stresses along the normalized length of the three paths (Figure 2, blue, red, and green lines) were calculated with the APDL algorithm and used as target training data. The disc part path (blue) was divided into twenty-one nodes, and the drums paths (green and red) were divided into six nodes.

## 3. Selection of the Training Method

The selection of the training method resulted from the literature review. It indicated that ANN is a tool capable of supporting or replacing classical numerical calculations. As a result of the preliminary numerical analysis presented in [10], the authors obtained knowledge in the field of optimal network selection. It provided knowledge on the network structure used. NARX (nonlinear autoregressive neural network with external input) and FeedForward neural networks with extended structure were tested. The NARX network was unable to solve the problem of profiling compressor disc and was therefore excluded from further analysis. All analysed neural networks had biases and sigmoidal activation function, except for the last layer, which included single neurons. The constructed network does not fully meet the assumptions of stress prediction accuracy in the optimisation task. Therefore, in the next stage of the work, numerical studies were carried out in order to improve the selection of the optimal ANN structure and training method. A comparative analysis of the selected methods was carried out. The following were compared:Scaled Conjugate Gradient (SCG) method [43];Momentum (GDM) method [44];Resilient back propagation algorithm (RPROP) [45].

The network was trained from one to four times. The data used to train the ANN and compare the results were the stresses that were generated based on the algorithm in the APDL language. The network was trained until reaching the fiftieth iteration of the training algorithm without improving the weight values.

Network calculations were performed using one to four hidden layers and 25 or 50 neurons in each layer. A total of 150 results were obtained from each network, for which the absolute value of the average and the largest relative error between the stresses determined with the ANN and the APDL algorithm were determined. The results of the analyses are presented in Figure 7, Figure 8 and Figure 9, where the “out” data are the ANN training database, and the “TEST” database is the newly drawn data, previously unknown to the network.

Conclusions from the research on selection of the network training:Almost all networks achieve higher errors for networks with 50 neurons in the layer;Multiple training of the network has the greatest impact on the quality of the RPROP algorithm results;The RPROP algorithm obtains relatively large discrepancies in the calculation accuracy depending on the number of hidden layers, and especially depending on their parity;Errors for the first iteration of the training process for networks with one hidden layer for the SCG and RPROP algorithm are similar;SCG and GDM algorithms have a rapid error drop for four-layer networks;GDM algorithm has low sensitivity to the number of hidden layers (except for the four-layer network) and the repetition of the training process.

The above conclusions provided information that no algorithm stands out in terms of quality. In further research, the use of any of the algorithms was not ruled out. Therefore, an attempt was made to minimise the error through optimisation of the ANN structure. As a result of the analysis, there was no need to repeat the network training process.

## 4. Selection of an ANN Structure

In order to determine the optimal ANN structure, two-criteria optimisation was performed with the use of a genetic algorithm. The influence of the number of hidden layers and neurons in a layer on the absolute value of the mean relative error and standard deviation was studied. Optimisation for three training algorithms (RPROP, SCG, and GDM) was performed, as tested and described in the previous chapter. The genetic algorithm performed calculations for 20 generations, simultaneously optimising 50 neural networks with a given number of hidden layers. Networks of 1–3 hidden layers and 1–50 neurons in each layer were studied. Two cases of training data size were also compared: ±5% and ±10% relative to the variable value in relation to the reference disc. These variables are described in the second paragraph. The results of the optimisation process are presented below. These are the stresses calculated in the seventh node of the first path, calculated using the ANN and the APDL algorithm. For the remaining nodes, a similar correlation to the results obtained for the seventh node was observed. The performed optimisation produced Pareto fronts. The results are presented:For the GDM algorithm, in Figure 10 and Figure 11;For the SCG algorithm, in Figure 12, Figure 13, Figure 14 and Figure 15;For the RPROP algorithm, in Figure 16, Figure 17, Figure 18 and Figure 19.

Based on the error distribution and standard deviation, it was found that the GDM algorithm is unable to perform the task regardless of the network structure. The sensitivity of the algorithm was observed only with respect to the span of the database. Figure 11 clearly shows two bands corresponding to the database span of 10% and 5%. During optimisation, a lack of sensitivity to the training accuracy was noticed in relation to the number of hidden layers and the number of neurons in the layer.

Figure 12, Figure 13, Figure 14, Figure 15, Figure 16, Figure 17, Figure 18 and Figure 19 present the Pareto fronts for selected neural networks for two variables (absolute value of the average relative error δ and its standard deviation σ). As a result of the research, it was noticed that the increase in the number of hidden layers reduces the span impact of the training database on the accuracy of the results. However, the more layers the network has, the lower the accuracy. This may be caused by an insufficiently large sample of learning data; however, for a single-layer network, the learning accuracy is satisfactorily high. For both training methods (SCG and RPROP), the Pareto front points for a neural network with one hidden layer at the data span have the smallest relative error values. In the case of the RPROP algorithm for networks with one and two hidden layers, for data ±5%, the Pareto fronts are arranged parallel to each other, creating bands of similar width, with a smaller average error of the network with a single hidden layer. For the SCG algorithm, the front band for a single-layer network looks similar to a network trained with the RPROP algorithm, but the band for a two-layer network is much wider (Figure 12 and Figure 16). Its centre is closer to the single-layer network band, and its minimum values correspond to its values. The reasons for the change in the width and position of the front can be sought in the different number of neurons in the layer, as shown in Table 1 and Table 2. The number of neurons in a given network are average values, rounded to integers, obtained in the optimisation process.

Table 2 summarises the averaged absolute values of the relative error and its standard deviation for the selected training algorithms and the number of neurons in the layer. The results were obtained for a network with one hidden layer, as this network has the lowest error rate compared to networks with more layers.

The RPROP algorithm for one hidden layer, excluding the network with two neurons, matches the training database worse than the SCG algorithm. The network trained with the SCG algorithm with two neurons in the hidden layer has the smallest relative error, and the network with two hidden neurons also trained with the SCG algorithm has the smallest standard deviation. Through the Pareto front, distinguished for a particular number of neurons in a layer by the combination of δ and σ, on the basis of the trend line, a qualitatively average ANN match to the selected database can be determined, as shown in Figure 20 and Figure 21.

The determination coefficient ($R^{2}$) for the SCG and RPROP algorithm with one hidden layer and from 2 to 4 neurons for the database ±5% was also compared. Figure 22, Figure 23 and Figure 24 present the training accuracy of selected algorithms for the best-matched neural networks. It is, respectively, $R^{2}=0.9978$ and $R^{2}=0.9725$ for the SCG (NN 20-2-1) and RPROP (NN 20-3-1) algorithms. The accuracy of the optimal ANN structure was calculated on the basis of a newly drawn dataset which was not used in the network training process.

Based on the above analysis, according to the Pareto approach, a neural network with one hidden layer and two neurons was selected, trained with the SCG algorithm using the database span of ±5% as optimal for a given task. The networks of three and four neurons were rejected because they have few points significantly deviating from the mean, which increases the standard deviation.

The presented algorithm allows the determination of von Mises stress values which can be part of the penalty function for further mass optimization of the structure. The model makes it possible to determine the stress distribution in a drum–disc structure if the mass-geometric properties of the blades are known. The biggest advantage of the algorithm is its short running time, at the cost of an error within the error limit of the FEM method. The algorithm has been tested on the example of a stress-distribution analysis of a real structure of a 14-stage jet engine compressor (Figure 4) and had an error comparable to error for a single stage of disc-drum structure (Figure 1). The algorithm is not able to determine the stress distribution in a design where the disc is not symmetrical or where the disc is connected directly to the shaft. Moreover, with variable disc, hub, or drum thickness, the algorithm fails. An additional limitation is that the effect of temperature in the structure is neglected. However, it is possible to easily modify the APDL algorithm and add one more input parameter to the ANN to include the effect of temperature on the stress distribution.

## 5. Conclusions

The analysis of the literature and the results of calculations showed that it is possible to use artificial neural networks to solve scientific problems in the area of rotating rotor machine elements, such as an aircraft turbine engine. Although papers [23,46,47] deal with the problems of implementing Artificial Neural Networks to solve engineering problems, the analyses are limited to flows or to the problem of frictional contact with elements loaded with uniform forces. There are no attempts to use SSNs for strength calculations of rotating machine elements. In our paper, we applied ANN to predict stress distribution due to reduce time of analysis and potentially optimization. Three ANN training algorithms (GDM, SCG, and RPROP) were tested. The analysis of the results showed that the algorithm with the momentum method (GDM) had the worst match to both the training and test data. Repeating the network training process did not significantly affect the qualitative results. The calculations of the compressor unit showed that the conjugate gradient (SCG) algorithm and the resilient backpropagation (RPROP) algorithm have the highest and comparable accuracy of calculations.

The concept of using the combination of APDL and the MATLAB language as a stress-based penalty function for compressor unit optimisation has been presented. The optimisation of network structures with the use of a genetic algorithm enabled the determination of the Pareto fronts for optimal network structures. These fronts were grouped depending on the training method and the number of neurons in the layer. This made it possible, according to the Pareto approach, to select a network with an optimal structure and training method. Thanks to the optimisation of the network structure using a genetic algorithm, it is possible to obtain a high regression coefficient with a small number of neurons and hidden layers in artificial neural networks.

The combination of multi-criteria optimisation in the Pareto approach used in the study for the prediction of stresses in order to minimise the mass of the aircraft engine subassembly structure can be considered as intentional. Pareto multi-criteria optimisation is a useful tool to determine the optimal structure of the Artificial Neural Network, while minimising both its mean error and standard deviation.

Further research should concentrate on improving the learning method of Artificial Neural networks with newer algorithms. For example, in papers [48,49] based on the Lyapunov–Krasovskii functional, reciprocally convex technique, Jensen’s inequality reduced the communication load in the network. An event-triggering scheme filtering has been proposed for delayed neural networks with sampled data and some suitable sufficient conditions of the considered neural networks were achieved. 

## Figures and Tables

**Figure 1 materials-15-04451-f001:**
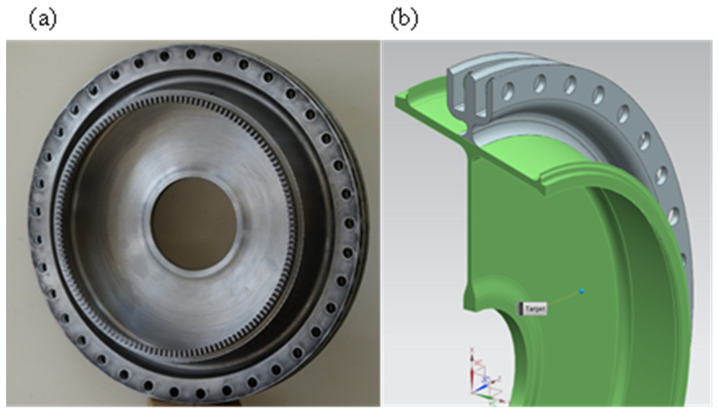
Optimised compressor disc: (**a**) real object, (**b**) CAD model.

**Figure 2 materials-15-04451-f002:**
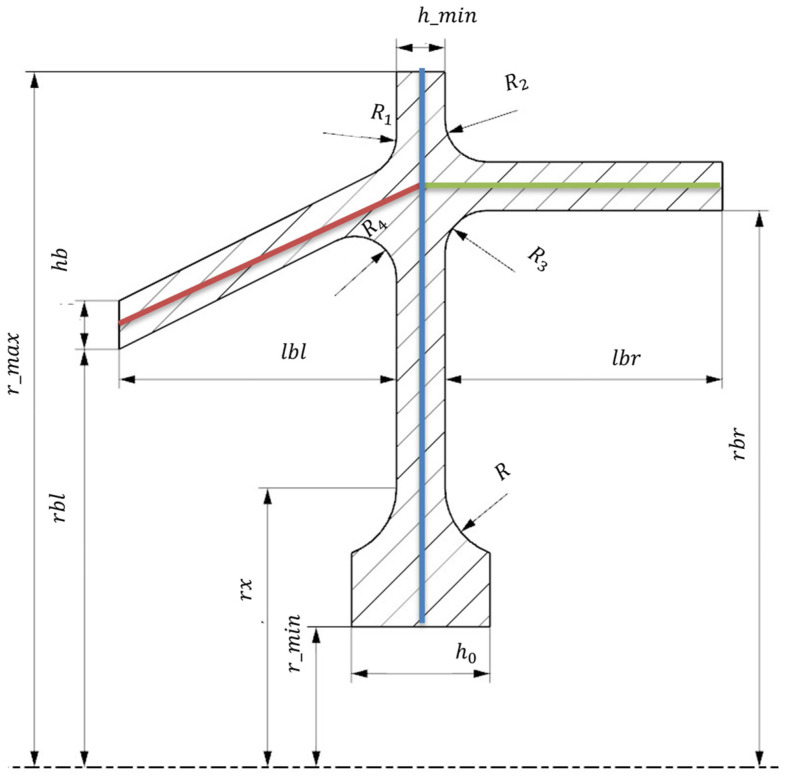
Parametric model of the disc with indication of the main dimensions. Colours indicate the paths along which the stresses for optimisation were determined.

**Figure 3 materials-15-04451-f003:**
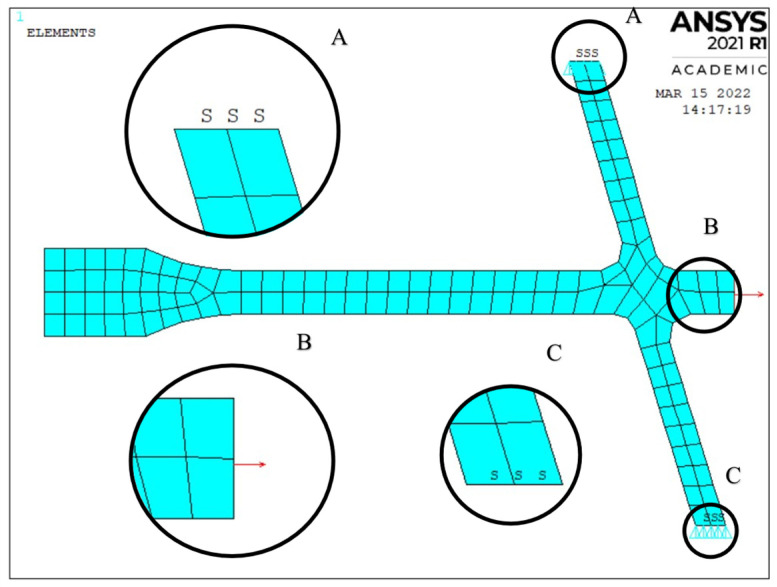
Discretised disc model with marked boundary conditions, where: A and C—symmetry condition zone, B—rim pull zone.

**Figure 4 materials-15-04451-f004:**
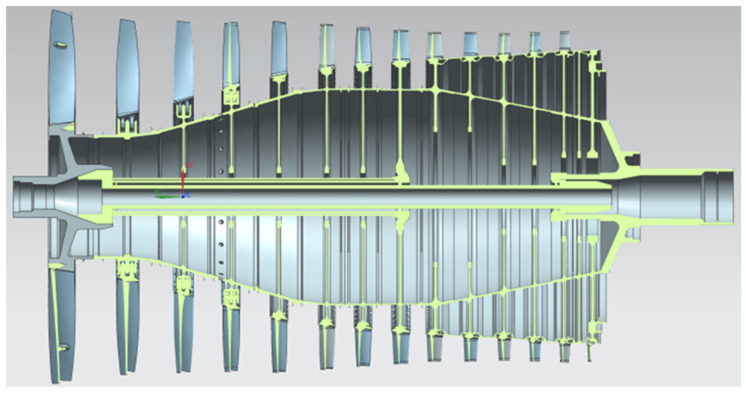
Compressor optimisation rotor.

**Figure 5 materials-15-04451-f005:**
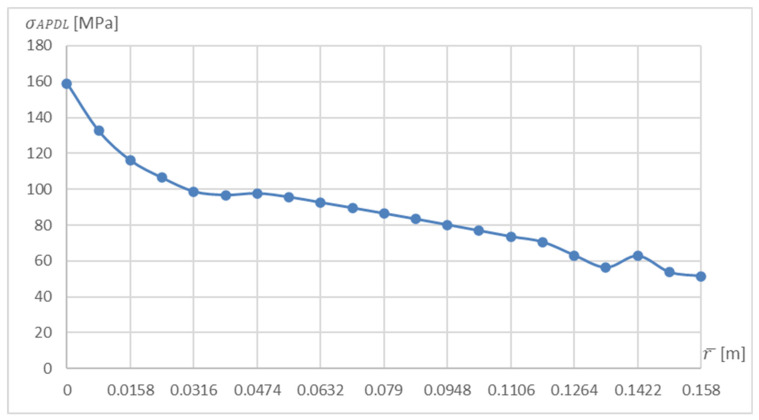
Stress distribution along the first path.

**Figure 6 materials-15-04451-f006:**
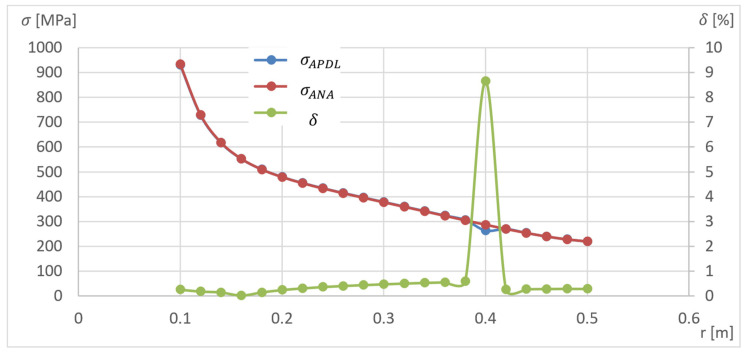
Results of program validation in APDL to determine the stress distribution in the disc.

**Figure 7 materials-15-04451-f007:**
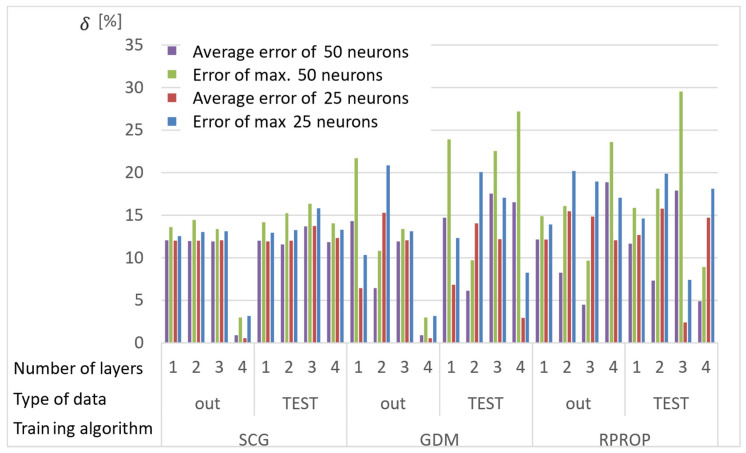
Maximum and average relative error for selected structures of neural networks and training algorithms after the first training cycle.

**Figure 8 materials-15-04451-f008:**
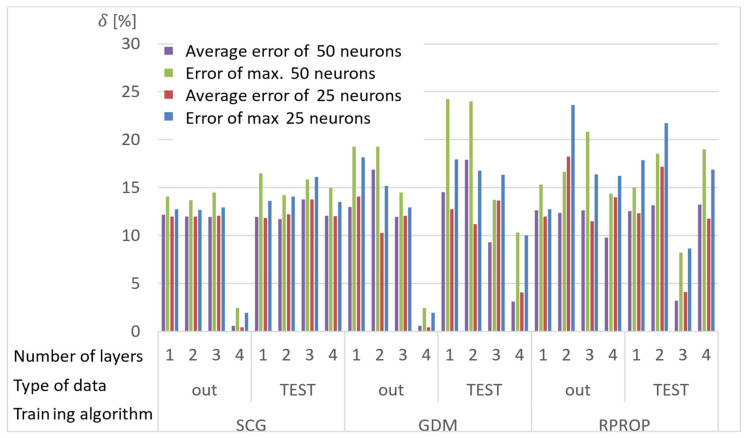
Maximum and average relative error for selected structures of neural networks and training algorithms after the second training cycle.

**Figure 9 materials-15-04451-f009:**
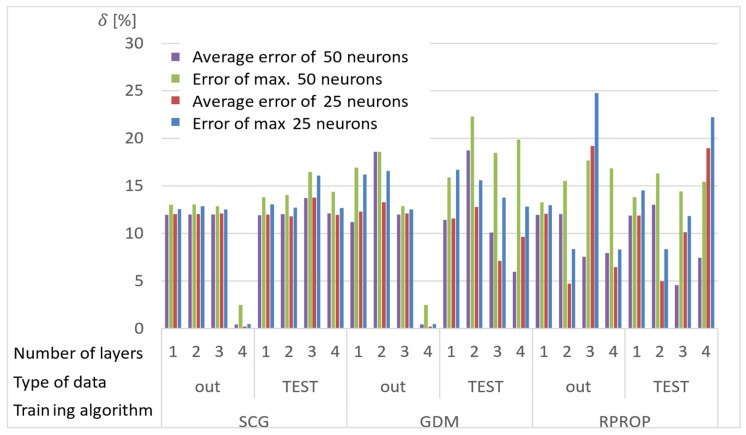
Maximum and average relative error for selected structures of neural networks and training algorithms after the third training cycle.

**Figure 10 materials-15-04451-f010:**
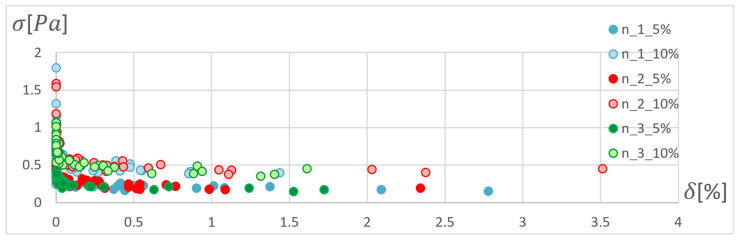
Pareto fronts for all networks trained using the GDM algorithm.

**Figure 11 materials-15-04451-f011:**
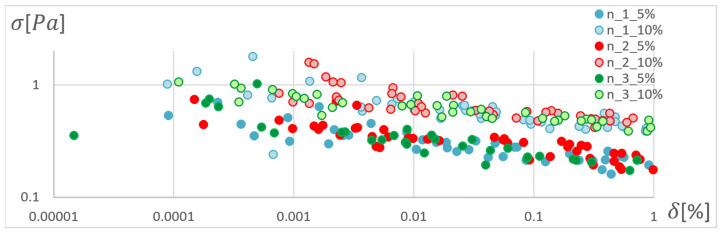
Pareto fronts for all networks trained using the GDM algorithm, in the zone of highest accuracy.

**Figure 12 materials-15-04451-f012:**
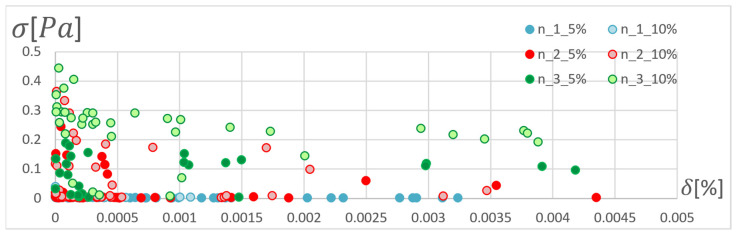
Pareto fronts for all networks trained using the SCG algorithm.

**Figure 13 materials-15-04451-f013:**
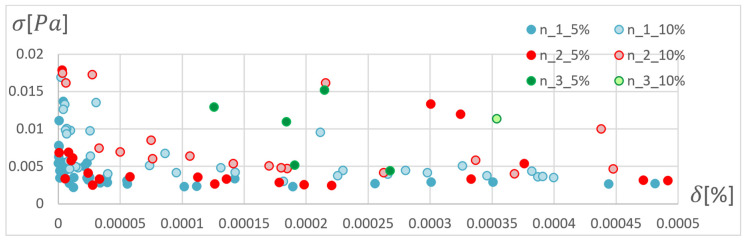
Pareto fronts representing the neural networks with the highest accuracy, trained using the SCG algorithm.

**Figure 14 materials-15-04451-f014:**
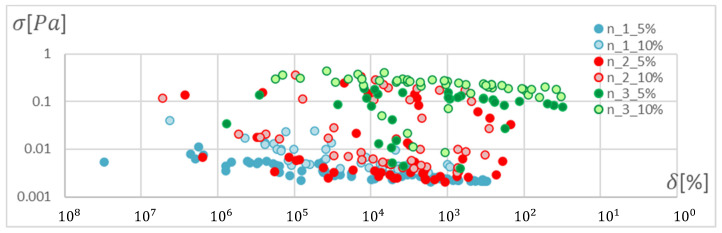
Pareto fronts presented on a logarithmic scale for all networks trained using the SCG algorithm.

**Figure 15 materials-15-04451-f015:**
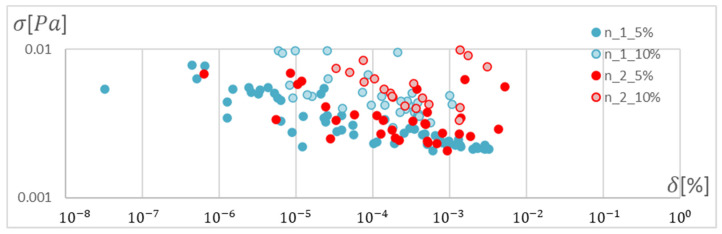
Pareto fronts for the two best-matched networks trained using the SCG algorithm.

**Figure 16 materials-15-04451-f016:**
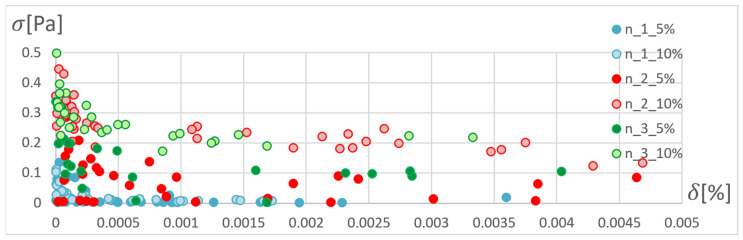
Pareto front for all studied networks trained using the RPROP algorithm.

**Figure 17 materials-15-04451-f017:**
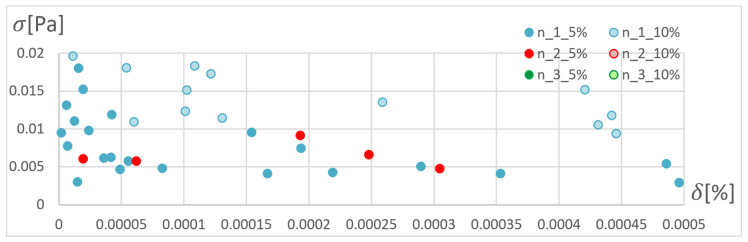
Pareto fronts representing the neural networks with the highest accuracy, trained using the RPROP algorithm.

**Figure 18 materials-15-04451-f018:**
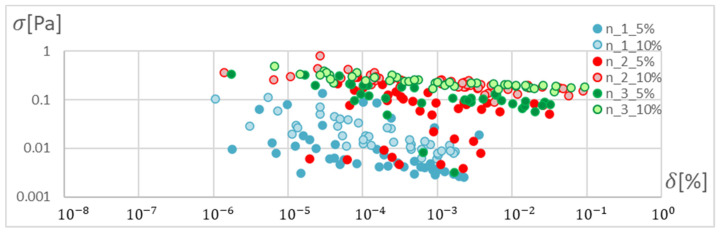
Pareto fronts presented on a logarithmic scalefor all studied networks trained using the RPROP algorithm.

**Figure 19 materials-15-04451-f019:**
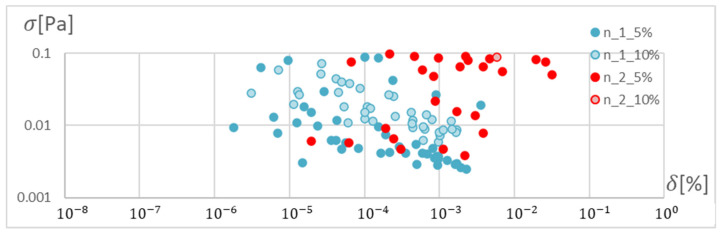
Pareto front for the two best-matched networks trained using the RPROP algorithm.

**Figure 20 materials-15-04451-f020:**
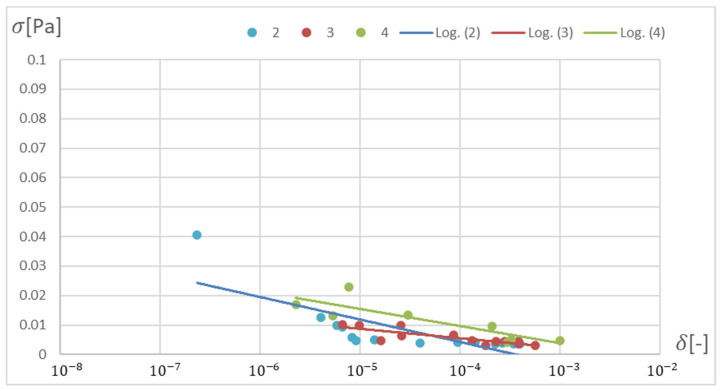
Pareto front depending on the number of neurons in the hidden layer with trend lines marked. Network trained with the SCG method.

**Figure 21 materials-15-04451-f021:**
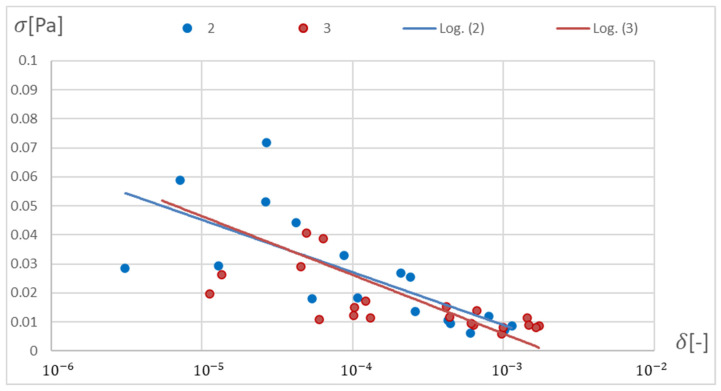
Pareto front depending on the number of neurons in the hidden layer with trend lines marked. Network trained with the RPROP method.

**Figure 22 materials-15-04451-f022:**
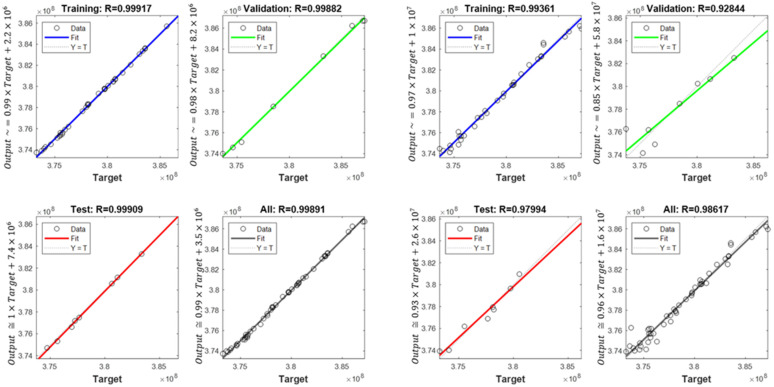
Matching the training data for the SCG algorithm (NN 20-2-1) on the left and RPROP algorithm (NN 20-3-1) on the right.

**Figure 23 materials-15-04451-f023:**
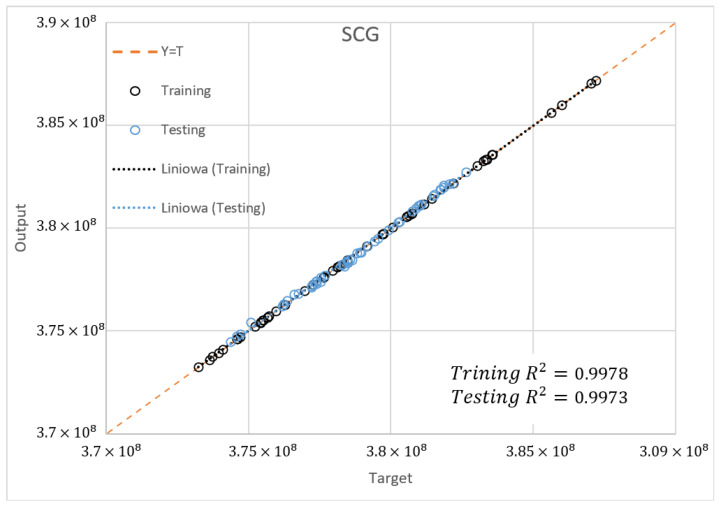
Comparison of data matching used for training and validation for the SCG algorithm (NN 20-2-1).

**Figure 24 materials-15-04451-f024:**
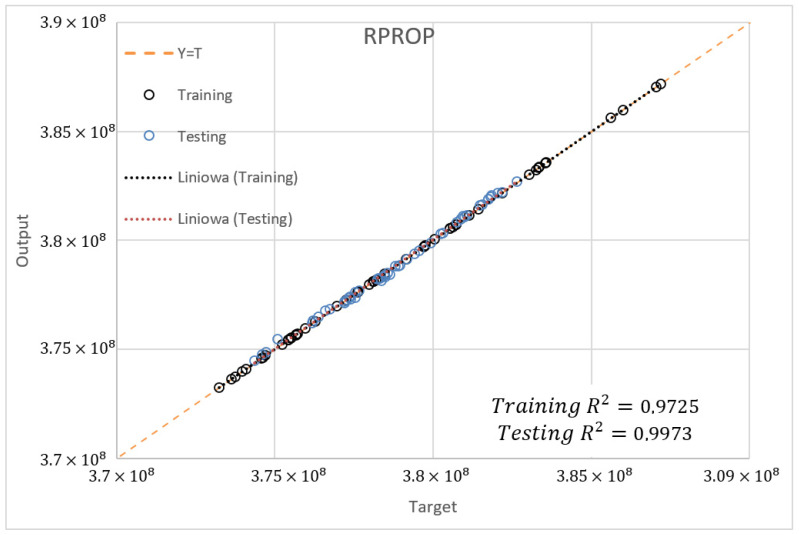
Comparison of data matching used for training and validation for the RPROP algorithm (NN 20-3-1).

**Table 1 materials-15-04451-t001:** Comparison of the optimal number of neurons in the hidden layer for selected training algorithms and different database spans.

Number of Hidden Layers		1		2				3					
Database span		5%	10%	5%		10%		5%			10%		
Average number of neurons in a layer	SCG	2	3	5	8	4	7	23	27	8	21	19	5
	RPTOP	3	3	12	6	16	6	17	22	9	33	23	6

**Table 2 materials-15-04451-t002:** Absolute value of the mean relative error δ and its standard deviation σ for selected training algorithms and the number of neurons in one hidden layer.

	SCG		RPROP	
Number of Neurons	δ [−]	σ [Pa]	δ [−]	σ [Pa]
2	1.30 × 10^−4^	1.30 × 10^−4^	3.07 × 10^−4^	2.53 × 10^−2^
3	1.79 × 10^−4^	1.79 × 10^−4^	5.34 × 10^−4^	5.34 × 10^−4^
4	3.30 × 10^−4^	3.30 × 10^−4^	-	-

## Data Availability

The data presented in this study are available on request from the corresponding author.

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
