# Peer review of "Artificial Neural Network Structure Optimisation in the Pareto Approach on the Example of Stress Prediction in the Disk-Drum Structure of an Axial Compressor"

_materials, 2022, doi:10.3390/ma15134451_

Round 1

Reviewer 1 Report

Dear Authors,

In this work, the authors developed, employed, and evaluated an Artificial Neural Network structure optimization in the Pareto approach on the example of stress prediction in the disk-drum structure of an axial compressor. The proposed method for disign a neural network to determine the optimal form of the disk-drum structures of an axial compressor makes it possible to reduce the complexity of calculating internal stresses under operating conditions.

The paper is physically sound, technically solid, and can be accepted for publication after the following minor issues are addressed:

1. Please delete the empty line 30.

2. In the Introduction, the areas of application of neural networks in the design of various products are considered, however the issue of determining weights in a neural network is not considered in sufficient detail. It is recommended to argue this process with actual examples. It is also recommended to add review material not related to the methods of creating neural networks and qualitatively compare the effectiveness of design methods based on different approaches.

3. Lines 116, 134, 146, 173, 226, 248, please correct the automatic reference to the figures.

4. Line 153-160, please justify the choice of neural network type.

5. Please, justify the loss of accuracy of the network with an increase in the number of hidden layers in terms of the design features of the product.

6. Please add the design features of the disk-drum structure of an axial compressor to the data filling technique in the neural network.

7. It is necessary to clearly define what technical parameters of the product are improved as a result of the use of a neural network in comparison with existing methods. It is also necessary to justify the scalability of the method to different operating conditions of the product.

8. Please indicate the limit conditions under which the proposed method ensures the reliability of the results.

Author Response

Dear Reviewer

First of all, I would like to thank you for your time and careful evaluation of our work. I would also like to thank you for all your substantial comments, which are extremely valuable to us. We have modified the manuscript according to your comments below and marked all changes with a highlighter. In the attachment, we also send the article with corrections to the your issues. We hope this is sufficient for the manuscript to be accepted for publication in Materials

On behalf of all the co-authors

Yours faithfully,

Rafał Kieszek

Dear Authors,

In this work, the authors developed, employed, and evaluated an Artificial Neural Network structure optimization in the Pareto approach on the example of stress prediction in the disk-drum structure of an axial compressor. The proposed method for disign a neural network to determine the optimal form of the disk-drum structures of an axial compressor makes it possible to reduce the complexity of calculating internal stresses under operating conditions.

The paper is physically sound, technically solid, and can be accepted for publication after the following minor issues are addressed:

  1. Please delete the empty line 30.

Line 30 has been deleted.

  1. In the Introduction, the areas of application of neural networks in the design of various products are considered, however the issue of determining weights in a neural network is not considered in sufficient detail. It is recommended to argue this process with actual examples. It is also recommended to add review material not related to the methods of creating neural networks and qualitatively compare the effectiveness of design methods based on different approaches.

The determination of weights in a neural network is described in lines 25-52. W. McCulloch and W. Pitts [1], developed the first formal description of the artificial neuron model in 1943, presented in paper [1]. The mathematical model of the McCulloch-Pitt neuron [1] is described in terms of the equation (1). The problem was develop in papers [2] and [3],

  1. b) It is recommended to argue this process with actual examples. It is also recommended to add review material not related to the methods of creating neural networks and qualitatively compare the effectiveness of design methods based on different approaches.

The main aim of paper was to reliably present the process of selecting the network structure using an example not presented in the literature. Inclusion of other methods would significantly increase the volume of the article

  1. Lines 116, 134, 146, 173, 226, 248, please correct the automatic reference to the figures.

Automatic reference to the figures has been removed.

  1. Line 153-160, please justify the choice of neural network type.

Justified in the lines 203-206. NARX (nonlinear autoregressive neural network with external input) and FeedForward neural networks with extended structure were tested. The NARX network was unable to solve the problem of profiling compressor disc and was therefore excluded from further analysis

  1. Please, justify the loss of accuracy of the network with an increase in the number of hidden layers in terms of the design features of the product.

Justified in the lines 301-303. This may be caused by an insufficiently large sample of learning data, however for a single-layer network the learning accuracy is satisfactorily high.

  1. Please add the design features of the disk-drum structure of an axial compressor to the data filling technique in the neural network.

Constructional features of the model are described in lines 185-196. In the paper, APDL algorithm is being used due to generate input training parameters described on Figure 2: r_max – radius of disk, r_min – radius of central hole, h0 - hub width, hmin - disc width, lbl – length of left drum, rbl – radius of left drum, lbr - length of left drum, rbr - radius of left drum, rb – radius of location of the center of the circle round the disc-hub joint. Additionally there are few parameters connected which material properties and blade geometry such as: Young module, density, Poisson’s ratio, mass of single blade, number of blades, radius of the center of mass of the lock, radius of the center of mass of the blade, volume of lock and rotational speed. The Von Mises stresses along the normalized length of the three paths (Figure 2, blue, red and green lines) were calculated by the APDL algorithm and used as target training data. The disc part path (blue) was divided into twenty-one nodes and the drums paths (green and red) were divided into six nodes.

  1. It is necessary to clearly define what technical parameters of the product are improved as a result of the use of a neural network in comparison with existing methods. It is also necessary to justify the scalability of the method to different operating conditions of the product.

Justified in the lines 185-192. In further part of paper, APDL algorithm is being used due to generate input training parameters described on Figure 2: r_max – radius of disk, r_min – radius of central hole, h0 - hub width, hmin - disc width, lbl – length of left drum, rbl – radius of left drum, lbr - length of left drum, rbr - radius of left drum, rb – radius of location of the centre of the circle round the disc-hub joint. Additionally, there are few parameters connected with material properties and blade geometry such as: Young module, density, Poisson’s ratio, mass of single blade, number of blades, radius of the centre of mass of the lock, radius of the centre of mass of the blade, volume of lock and rotational speed. The Von Mises stresses along the normalized length of the three paths (Figure 2, blue, red and green lines) were calculated by the APDL algorithm and used as target training data. The disc part path (blue) was divided into twenty-one nodes and the drums paths (green and red) were divided into six nodes.

  1. Please indicate the limit conditions under which the proposed method ensures the reliability of the results.

Limits are described in lines 360-372. The presented algorithm allows the determination of von Mises stress values which can be part of the penalty function for further mass optimization of the structure. The model makes it possible to determine the stress distribution in a drum-disc structure if the mass-geometric properties of the blades are known. The biggest advantage of the algorithm is its short running time, at the cost of an error within the error limit of the FEM method. The algorithm has been tested on the example of stress distribution analysis in a real structure of 14-stage jet engine compressor and had comparable error for stages of a typical disc-drum structure (Figure 1). The algorithm is not able to determine the stress distribution in a design where the disc is not symmetrical or where the disc is connected directly to the shaft. Also, for variable disc, hub or drum thickness, the algorithm fails. An additional limitation is that the effect of temperature in the structure is neglected. However, it is possible to easily modify the APDL algorithm and add one more input parameters to ANN to include the effect of temperature on the stress distribution.

Reviewer 2 Report

I have a series of very major concerns on this paper.

1) Is it possible to apply this modeled problem to time delayed system, need detailed discussion.

2) Motivations and applications of stress distribution in a disk-drum structure compressor 12 stage of an aircraft turbine engine are not clear. What are they modeling? Where are they applied? Which systems do they model?

3) What kind of strategy are considered in these networks? Can they be found in the market?

4) Some sentence throughout the paper are strange which make the paper lack readability. Author should check and correct it.

5) How to construct the derivation (1), need detailed mathematical derivations or give the proper citation.

6) The novelty of the paper should be highlighted, especially when compared with some new publications.

7) Discuss the technical difficulty in dealing with proposed scheme.

8) According to the topic of the paper, the authors should discuss some interesting problem in the introduction section and cite them, such as  Event-triggered H∞ filtering for delayed neural networks via sampled-data; Global exponential stability of fractional order complex-valued neural networks with leakage delay and mixed time varying delays.

9) Some comparisons needed to show the effectiveness of the work.

Author Response

Dear Reviewer

First of all, I would like to thank you for your time and careful evaluation of our work. I would also like to thank you for all your substantial comments, which are extremely valuable to us. We have modified the manuscript according to your comments below and marked all changes with a highlighter. Please find attached an article with corrections that answer your concerns. We hope this is sufficient for the manuscript to be accepted for publication in Materials

On behalf of all the co-authors

Yours faithfully,

Rafał Kieszek

Open Review

English language and style

( ) Extensive editing of English language and style required
(x) Moderate English changes required
( ) English language and style are fine/minor spell check required
( ) I don't feel qualified to judge about the English language and style

Comments and Suggestions for Authors

I have a series of very major concerns on this paper.

1) Is it possible to apply this modeled problem to time delayed system, need detailed discussion.

Algorithm can not be applied to time delayed system, because APDL algorithm based on MES is too slow. The single case analysis with the proposed neural network is baseless because it can be done with single FEM analysis. Algorithm is useful as part of aim function in optimization (for example based on stresses penalty function). For a neural network, an FEM analysis must be performed anyway, which is sufficient to determine the stress distribution for a single case. In our university, research was carried out as part of a PhD thesis, on the topic "Development and research of advanced algorithms for controlling the magnetic bearing of a jet engine", where an attempt was made to use artificial neural networks to control the magnetic bearing. In the research associated with the PhD thesis, it was shown that the time delays there were too large for the use of neural networks. In our work, the timing of ANNs is much time consuming than in the Ph.D. thesis.

2) Motivations and applications of stress distribution in a disk-drum structure compressor 12 stage of an aircraft turbine engine are not clear. What are they modeling? Where are they applied? Which systems do they model?

Described in lines 13-14 and 360-372. The presented algorithm allows the determination of von Mises stress values which can be part of the penalty function for further mass optimization of the structure. Von Mises stress values can be part of the penalty function for further mass optimization of the structure. The model makes it possible to determine the stress distribution in a drum-disc structure if the mass-geometric properties of the blades are known. The biggest advantage of the algorithm is its short running time, at the cost of an error within the error limit of the FEM method. The algorithm has been tested on the example of stress distribution analysis in a real structure of 14-stage jet engine compressor and had comparable error for stages of a typical disc-drum structure (Figure 1). The algorithm is not able to determine the stress distribution in a design where the disc is not symmetrical or where the disc is connected directly to the shaft. Also, for variable disc, hub or drum thickness, the algorithm fails. An additional limitation is that the effect of temperature in the structure is neglected. However, it is possible to easily modify the APDL algorithm and add one more input parameters to ANN to include the effect of temperature on the stress distribution.

Moreover, analysis of stress distribution is a standard procedure for strength analysis of any drum-disc system, including multistage compressors, which is consistent with the literature. The choice of the 14-stage compressor unit was related to the attempt to undertake the analysis of a complex structural system, with the possibility of access to the real object and its technical documentation. The system of disks and drums, which are the main components of an axial compressor, was modelled in terms of geometric dimensions, having a significant impact on the mass of the whole object. The geometry of the structural system is modeled to achieve the criterion of minimum mass while maintaining the structural strength obtained from the stress distribution. In our approach, the system is the entire aircraft turbine engine, which consists of the inlet, compressor, combustion chamber, turbine, and exhaust nozzle. All subsystems can be modeled. In this case, the authors have covered only one major problem, which is the design layout of the axial compressor. Each of the subsystems has additionally individual issues to solve.

3) What kind of strategy are considered in these networks? Can they be found in the market?

Described in lines 206-207. All analyzed neural networks had biases and sigmoidal activation function, except last layer which included single neurons. As a result of our analysis, we chose 20-2-1 FeedForward artificial neural network as optimal.

We can not be sure about implication of any architecture of neural network in the market, because in aviation, it is usually technical data which are not shared with public eye by business.

4) Some sentence throughout the paper are strange which make the paper lack readability. Author should check and correct it.

A request for language proofreading was sent to the editor.

5) How to construct the derivation (1), need detailed mathematical derivations or give the proper citation.

Added the source of equation (1) and described all variables (lines 145-150), based on paper [41]:

Ft – centrifugal force from the lock,

Fz - centrifugal force from the blade,

n - number of blades,

rmax - outer radius of the disc,

hmin - thickness of the disc.

6) The novelty of the paper should be highlighted, especially when compared with some new publications.

There are no works using artificial neural networks to predict the stress distribution in disc and drum structures. Moreover, the determination of the neural network structure is usually neglected by the authors. Our aim was to reliably present the process of selecting the network structure using an example not presented in the literature.

Aviation companies involved in the design of aircraft engines do not publish material on the use of this type of analysis. Our experience in dealing with such companies is that they are interested in this topic.

7) Discuss the technical difficulty in dealing with proposed scheme.

Described in lines 360-372. The presented algorithm allows the determination of von Mises stress values which can be part of the penalty function for further mass optimization of the structure. The model makes it possible to determine the stress distribution in a drum-disc structure if the mass-geometric properties of the blades are known. The biggest advantage of the algorithm is its short running time, at the cost of an error within the error limit of the FEM method. The algorithm has been tested on the example of stress distribution analysis in a real structure of 14-stage jet engine compressor and had comparable error for stages of a typical disc-drum structure (Figure 1). The algorithm is not able to determine the stress distribution in a design where the disc is not symmetrical or where the disc is connected directly to the shaft. Also, for variable disc, hub or drum thickness, the algorithm fails. An additional limitation is that the effect of temperature in the structure is neglected. However, it is possible to easily modify the APDL algorithm and add one more input parameters to ANN to include the effect of temperature on the stress distribution.

8) According to the topic of the paper, the authors should discuss some interesting problem in the introduction section and cite them, such as  Event-triggered H∞ filtering for delayed neural networks via sampled-data; Global exponential stability of fractional order complex-valued neural networks with leakage delay and mixed time varying delays.

Thank you for bringing these issues to our attention. Due to the limited size of the paper, the focus is on the computational side rather than the theoretical side. We will take this into consideration in our further work.

9) Some comparisons needed to show the effectiveness of the work.

In one of our previous papers [10], we describe time comparison analysis of single genetic algorithm and genetic algorithm supported by artificial neural networks. In his master thesis, one of the authors used genetic algorithm to optimization compressor disc. However time of that analysis was nearly one week. In this paper time analysis was not considered, because our research trial was covered just one engineering problem. In further papers we are going to perform statistical time consuming and efficacy analysis of algorithm for other engineering problems.

Round 2

Reviewer 2 Report

Authors should consider the following comments:

1. According to the topic of the paper, the authors should discuss some interesting problems in the conclusion section as future direction and cite them, such as  Event-triggered H∞ filtering for delayed neural networks via sampled data; Global exponential stability of fractional-order complex-valued neural networks with leakage delay and mixed time-varying delays.

2. The novelty of the paper should be highlighted, especially when compared with some new publications

3. In equ (5) need mathematical derivations. Authors should give more explanation in the responses.

Author Response

Dear Reviewer

Yours faithfully,

Rafał Kieszek
